# Inter-Unit Consistency and Validity of 10-Hz GNSS Units in Straight-Line Sprint Running

**DOI:** 10.3390/s22051888

**Published:** 2022-02-28

**Authors:** Amandeep Kaur Chahal, Jolene Ziyuan Lim, Jing-Wen Pan, Pui Wah Kong

**Affiliations:** Physical Education and Sports Science Academic Group, National Institute of Education, Nanyang Technological University, Singapore 637616, Singapore; amandeep001@e.ntu.edu.sg (A.K.C.); nie20.lzj@e.ntu.edu.sg (J.Z.L.); nie173748@e.ntu.edu.sg (J.-W.P.)

**Keywords:** Global Navigation Satellite System, reliability, distance, speed, video, movement analysis

## Abstract

The present study aimed to investigate the inter-unit consistency and validity of multiple 10-Hz Catapult Global Navigation Satellite System (GNSS) units in measuring straight-line sprint distances and speeds. A total of 13 participants performed one 45.72-m linear sprint at maximum effort while wearing all eight GNSS units at once. Total run distance and peak speed recorded using GNSS units during the sprint duration were extracted for analysis. Sprint time and peak speed were also obtained from video recordings as reference values. Inter-unit consistency was assessed using intraclass correlation coefficients (ICC) and standard errors of measurements (SEM). For a validity test, one-sample *t*-tests were performed to compare each GNSS unit’s distance with the known distance. Additionally, Wilcoxon signed-rank tests were performed to compare each unit’s peak speed with the reference peak speed measured using video analysis. Results showed poor inter-unit consistency for both distance (ICC = 0.131; SEM = 8.8 m) and speed (ICC = 0.323; SEM 1.3 m/s) measurements. For validity, most units recorded a total distance (44.50 m to 52.69 m) greater than the known distance of 45.72 m and a lower peak speed (7.25 (0.51) m/s) than the video-based reference values (7.78 (0.90) m/s). The present findings demonstrate that there exist variations in distance and speed measurements among different units of the same GNSS system during straight-line sprint running. Practitioners should be aware of the window of errors associated with GNSS measurements and interpret the results with caution. When making comparisons over a season, players should wear the same unit every time if logistically possible.

## 1. Introduction

In sports, the movement characteristics of players during competitions and training are of interest for in-game analyses. Traditionally, player activity data were manually collected on pen and paper, which was extremely labor-intensive and time-consuming [1,2]. With technological advancements in time-motion analysis, more convenient methods, such as video analysis, have been regularly used to track player movements during competitions and training. However, video analysis can be troublesome to set up and requires extensive manual analysis after data collection. The development of Global Navigation Satellite System (GNSS) and Global Positioning System (GPS) units, which are light, small, and portable, allows for simultaneous movement patterns analyses of multiple players [3,4]. Since then, the use of GNSS/GPS for athlete tracking has become widespread in various sports, such as soccer, rugby, and field hockey [3,5,6,7] due to the ease of data collection and quality of analysis provided by these systems [8,9,10,11]. Additionally, GNSS/GPS units are the conventional technology used for the assessment of external training load variables in team sports due to their ability to give real-time feedback. This is essential, given the limited amount of time to process data and carry out post-session analysis [3,12,13].

GPS is a navigation system based on connections to satellites that allows locations of users to be triangulated through signals sent out by the satellites and received by the units [14]. The accuracy of data is dependent on the configuration of the satellites in relation to the receiver and how evenly spaced they are, known as the dilution of precision (DOP). Triangulation of position is the most accurate when one satellite is directly overhead the receiver while the rest of the satellites are evenly spaced around the horizon (DOP = 1). It has generally been suggested that GPS units require at least 4 satellites for data to be considered accurate. In addition, satellites that are more evenly spaced are considered stronger than when satellites are close together [14]. GPS units are usually combined with microsensors such as accelerometers that are capable of recording movements in three planes, allowing the intensity of body load (also known as player load in some systems) to be measured. In addition, the inclusion of gyroscopes and magnetometers in these units allows for directional orientation and rotational velocity to also be measured [15]. Sampling rates of GPS units may range from 1 to 15 Hz, indicating the multiple speeds at which the GPS units collect data. Existing studies have shown that higher sampling rates increase the accuracy of performance indicators [16,17,18] recorded by the GPS units. For instance, 10-Hz GPS units are more accurate than those of lower sampling rates in measuring total distance covered during both linear activities and sport-specific circuits and measuring peak speed [19]. No additional accuracy has been found between 10-Hz and 15-Hz GPS units [19], indicating that a sampling rate of 10 Hz could be sufficient.

GNSS/GPS units provide a multitude of movement variables including distance, speed, acceleration/deceleration, and metabolic power [3,20]. These movements can be purposefully analyzed (external training load) to comprehend the positional demands in sports, allowing practitioners to design programs that accurately emulate and equip athletes for their specific sport [3]. Furthermore, the GNSS/GPS data have also been proven useful in aiding practitioners to understand physiological and technical demands of their players through the extraction of various external training load measures such as volume, intensity, and frequency [21]. Such information can inform and guide coaches and sport scientists to develop appropriate conditioning and recovery plans [22,23].

While GNSS/GPS units provide practical and useful feedback, environmental objects such as surrounding tall buildings [24], atmospheric pressure [25], as well as the level of satellite giving out the signals (with signals from lower satellites having to go through more atmosphere) can result in obstruction of signals, leading to lower signal-to-noise ratio and lower accuracy in measurements. Hence, it is important to establish the validity and reliability of these units before applying them in sports [4,16,26]. Testing validity provides an understanding of the differences between the measures recorded by the units and standard measures. Reliability testing, on the other hand, tests reproducibility of values when the same test is repeated by another unit. While studies have generally agreed that GNSS/GPS devices can be reliable in straight-line running, there is a sizable inconsistency in accuracy among the models of GNSS/GPS manufacturers [12,18,27]. Imparting the validation of one system to another can be imprecise even if it is introduced by the same manufacturer [28]. While Johnston and co-workers [29] used a different software to collect and analyze GPS data collected by other brands of GPS units, the authors cautioned that the mismatching in GPS models may have influenced the movement demand data. Hence, it is vital to carry out an independent and thorough trial for each new GNSS/GPS device (hardware) and its analysis tool (software).

Within the same system, high consistency between different GNSS/GPS units is critical especially in team sports whereby each player wears an independent unit. Previous studies examining the inter-unit reliability of GPS units have placed multiple units on solid objects such as a golf cart and motorcycle [26], plastic sled [30], and a trundle wheel [31]. It should be noted that the movement trajectories of solid objects may differ from those of the human players who can freely move individual body segments in different directions at various magnitudes. There are very few studies placing GPS units on human participants and these studies typically compared among only two to four units each time [27,32]. To the best of the authors’ knowledge, only one study has tested the inter-unit reliability of eight GPS units on an individual [33]. Although their study found inter- and intra-receiver reliability to be acceptable, the GPS units were sampled at 1 Hz which is far below the recommended frequency of 10 Hz for accurate measurements [19]. Thus, there is a need to examine the inter-unit consistency and validity of multiple GNSS/GPS units sampled at sufficiently high frequency (i.e., at least 10 Hz) with the units placed on human participants and not solid objects.

This study, therefore, aimed to investigate the inter-unit consistency and validity of 10-Hz Catapult GNSS (S5 OptimEye, Catapult Innovations, Melbourne, Australia) units during straight-line sprint running. Eight GNSS units were analyzed using the Sprint software developed by Catapult. It was hypothesized that all GNSS units, when placed on human participants, would be consistent and accurate in measuring distances and peak speeds during sprint running [18].

## 2. Materials and Methods

### 2.1. Participants

This study was approved by the Nanyang Technological University Institutional Review Board (IRB-2020-09-033). Thirteen active participants (4 males, 9 females) were recruited via convenient sampling [age 21.6 (1.6) years, height 170.6 (7.7) cm, body mass 63.1 (10.1) kg]. To be eligible for this study, participants must have been training with a sports team at least twice a week and had minimally a year of experience in the specified sport. Additionally, they were required to be injury-free and pain-free at the time of the study.

### 2.2. Equipment

The Catapult S5 OptimEye GNSS system was used in the present study. Accessing GPS and Global Navigation Satellite System (GLONASS) satellite constellations, this GNSS system ensures high-quality data even in challenging performance environments (https://www.catapultsports.com (accessed on 1 October 2020)). Eight 10-Hz GNSS units were worn at once on each participant during the test (Figure 1). The eight GNSS units were placed near the mid-back area using a custom-made strap, with slightly different positions. While the tightness of strap was adjusted to fit individual body sizes, the relative positions of the GNSS units on the strap remained consistent across all participants.

### 2.3. Experimental Protocol

This experiment involved one single visit to the field hockey pitch at the National Institute of Education, Nanyang Technological University, Singapore. GNSS data were collected outdoors without high surrounding buildings to enhance satellite reception [34]. After sufficient warm-up and putting on the strap with 8 GNSS units, participants were asked to perform one straight-line sprint across the hockey pitch with maximum effort (Figure 2). A sprint distance of 45.72 m was chosen as it is exactly half of the hockey pitch with a clear marked line. This distance also allowed sufficient time for participants to reach their peak speed. To help participants discern the start and end points, the sprint area was marked out using colored cones. Participants were asked to run pass the 45.72 m end-line before they could slow down. The sprint test was recorded using two video cameras at 60 Hz for subsequent determination of sprint times and peak speeds. To minimize the effect of camera lens distortion, we used two smartphone cameras to cover the entire 45.72 m range (Figure 2). The midline distance of 22.86 m was used to calibrate each camera (0 to 22.86 m, 22.86 to 45.72 m) as the midline can be clearly seen from both camera views.

GNSS units were switched on at least 5 min before the units were strapped on the participants. After strapping on all units, participants were verbally briefed and then asked to familiarize themselves with the task. The GNSS units were switched on for more than 15 min to receive the complete almanac before the commencement of the test. Participants were also instructed to stay still for 30 s, before the start of the sprint. This was to enable subsequent determination of the start time for each trial when the speed increased sharply from zero.

### 2.4. Data Processing

The GNSS movement data were downloaded using the manufacturer’s software (Catapult Sprint Version 5.1.7, Melbourne, Australia) at the default ‘GPS rate’ of 10 Hz. Customized MATLAB codes were written to extract the relevant distance and speed time-series data using MATLAB (R2021a, MathWorks, Natick, MA, USA). The start of the sprint was identified from a sharp and continuous increase in speed above a threshold of 0.5 m/s. The duration each participant took to complete the 45.72 m distance was obtained based on the video recordings of the sprint. This sprint duration was then used to determine the end time of the sprint in the GNSS data. From the start to the end of the sprint, total distance traveled, and peak speeds were obtained from each of the 8 GNSS units. Raw GNSS data were used without further down sampling, filtering, or smoothing procedures. Due to transmission and technical errors, it was not possible to obtain complete data sets from all 8 GNSS units throughout all trials. Among the 13 participants, 7 had complete data set and 6 had missing data from either 1 or 2 GNSS units.

For validity analysis, a reference value of the gold standard was needed. In the present study, the total distance was 45.72 m, which was the known size of half of a standard field hockey pitch. This distance was also confirmed by experimental measurement using a trundle wheel. To calculate the speed from position data, manual digitization of the player’s center of the head was performed through the sprint duration using the software Kinovea (version 0.9.3, Kinovea, Bordeaux, France, available for download at: http://www.kinovea.org (accessed on 15 April 2021)). The present study used video analysis as the gold standard for kinematics, which is aligned with previous work evaluating the accuracy of 10 Hz GPS system [12]. Kinovea has been demonstrated as a reliable and accurate tool for video-based angular and linear measurements via digitization of x- and y-axis coordinates [35]. While an optimal angle of 90° was recommended, an accepted level accuracy was also established when the camera was placed within an angle range of 45° to 90° [35].

Figure 3 illustrates examples of the speed-time data measured using one GNSS unit and video analysis. The raw speed data from videos were low-passed filter at 10 Hz to remove the noise associated with manual digitization. The peak value of the filtered speed data during the entire sprint duration was then identified. This video-based peak speed was used as a reference value in the subsequent validity analysis of GNSS units. The mean (SD) of the raw and filtered peak speeds were 7.82 (0.81) m/s and 7.78 (0.90) m/s, respectively.

### 2.5. Statistical Analyses

Statistical analyses were carried out on JASP (version 0.14.1, JASP Team 2020) and SPSS (version 26.0, IBM Corp., Armonk, NY, USA). Data are expressed as mean (standard deviation). An alpha level of *p* < 0.05 was set as the level of significance. Inter-unit consistency was assessed using intraclass correlation coefficients (ICC). ICC was interpreted as *slight* (<0.20), *fair* (0.21–0.40), *moderate* (0.41–60), *substantial* (0.61–0.80), or *almost perfect* reliability (>0.80) [36,37]. Standard error of measurement (SEM) was calculated from the ICC results using the formula: SEM=SD×√(1−ICC).

For the validity assessment, one-sample *t*-tests were performed to compare the distance measured using each GNSS unit with the known distance of 45.72 m. Effect sizes were indicated by Cohen’s *d* and interpreted as *small* (0.2 ≤ d < 0.5), *medium* (0.5 ≤ d < 0.8), or *large* (d ≥ 0.8). Since the speed data were not normally distributed, non-parametric statistical tests were employed. Specially, Wilcoxon signed-rank tests were used to compare each GNSS unit’s peak speed with the reference speed measured using video analysis. Effect size (r) for the Wilcoxon signed-rank tests was calculated from the Z-value and interpreted as *small* (0.1 ≤ |r| < 0.3), *medium* (0.3 ≤ |r| < 0.5), or *large* (|r| ≥ 0.5).

## 3. Results

### 3.1. Inter-Unit Consistency

The results of ICC analysis showed *slight* reliability for the total sprint distance and *fair* reliability for peak speed (Table 1). These results indicate that the 8 tested GNSS units are not sufficiently consistent among themselves.

### 3.2. Validity

Most GNSS units recorded a total distance greater than the known distance of 45.72 m (Table 2). While statistical significance was only found in two units, the effect sizes of the differences were large across all units. These results indicate that GNSS units, although belonging to the same system, do not always measure distance with the same degree of accuracy.

Compared with the reference speed data obtained from video analysis, Unit 4 measured significantly higher peak speed (*p* = 0.010, large effect size, Table 3). No significant differences were identified between other GNSS units and video analysis, with data of 4 units approaching statistical significance (Units 1, 3, 7, 8). In general, most GNSS units measured a lower peak speed (7.25 (0.51) m/s) than the video-based value (7.78 (0.90) m/s) and the effect sizes of the differences were medium to large.

## 4. Discussion

The aim of the study was to investigate the inter-unit reliability and validity of multiple 10-Hz Catapult GNSS units during straight-line sprint running. Inter-unit consistency was assessed among eight GNSS units worn on each participant, and validity was tested by comparing total distance and peak speed against criterion-referenced values. The most prevailing outcomes were that despite all GNSS units belonging to the same system, low inter-unit reliability and varied accuracies in distance and speed measurements were found during fast speed running.

### 4.1. Distance

We originally expect that all GNSS units, when placed on the participant, would be consistent and accurate in measuring total distance traveled during 45.72-m sprint. However, there was only *slight* reliability for inter-unit consistency among the eight GNSS units and two out of eight units (Units 1 and 4, Table 2) had significantly different values from the criterion distance. In addition, seven out of the eight GNSS units overestimated the values during the straight-line sprint. These results in the present study are somewhat in congruence with previous research which reported moderate errors when measuring total distance over very high-speed running (>5.56 m/s) [17]. Additionally, overestimation of the total distance measured using GNSS units has also been found when the sprinting distances were set as 15 m and 30 m [34,35]. The reliability and accuracy may also be affected by rapid changes in speed during the acceleration phase of the sprint. A previous study revealed that distance measures over the post-acceleration phase of 20–40 m were more accurate than the acceleration phase of 0–20 m in a 40-m linear acceleration run [16], suggesting that smaller variations in speed may facilitate more accurate measures in distance. In the present study, participants started from a stationary position and were asked to sprint as fast as they could using maximal effort. Hence, phases with great variations in speed could have resulted in inconsistent and less accurate total distances measurement across different units. It is also possible that some participants did not sprint in a perfectly straight line hence covering a longer distance than the reference value of 45.72 m. Although the deviation from a straight line can be expected to be quite small, this could partly explain why seven out of eight GNSS units recorded a longer total distance than the reference value based on the distance between standard marked lines on the field. Finally, the GNSS units could miss data owing to the poor satellite connection [19]. This may have caused measurement errors in certain GNSS units, leading to inconsistency among the different units.

### 4.2. Peak Speed

This study hypothesized that multiple GNSS units of the same system would be consistent and accurate in measuring peak speed during a maximal effort sprint. The results demonstrated *fair* reliability among the eight GNSS units and that seven out of the eight units generally measured lower peak speeds than that video-based reference values (Table 3). The results are not in line with previous findings which suggested confidence in 10-Hz GNSS units being able to accurately measure consistent speeds and velocities [18]. The discrepancy in the peak speeds measured can be attributed to the compromises when measuring instantaneous velocities during great decelerations [38] and accelerations [28]. Hence, rapid changes in speed during the acceleration phase of the 45.72-m sprint in the present study could affect the accuracy of the GNSS units when measuring peak speeds. Higher accuracy and inter-unit reliability may be expected if GNSS units are applied to measure speed during a stable phase with small decelerations or accelerations.

Compared with the video analysis which was used as the golden standard for the speed measurement, only one GNSS unit displayed statistically significant result (Unit 4, Table 3). It is worth noting that the effect sizes of the differences were medium to large across all units regardless of statistical significance. As the GNSS units tend to register lower peak speeds (7.25 (0.51) m/s) than video-based reference values (7.78 (0.90) m/s), such differences cannot be disregarded. Sport practitioners should keep in mind that GNSS readings may slightly underestimate peak speeds during high-speed running and interpret the results with consideration of the error window (SEM = 1.3 m/s).

### 4.3. Limitations

There were a few limitations to the current study. Firstly, six participants had missing data due to either faulty units or the poor connection to the satellites. The current sample size of 13 participants was smaller than expected since the experiment was halted prematurely due to the COVID-19 pandemic. A larger sample size may have brought about more reliable results, which was unfortunately not possible due to time constraints. Secondly, environmental factors (e.g., presence of clouds) during the experiment may have occurred and affected the results. Thirdly, we acknowledge that the use of smartphone cameras can reduce the accuracy of data collected due to optical effects, such as lens distortion and parallax error. For fast sprint movements, the relatively low frame rate of 60 Hz could have also compromised accuracy of speed and time data collected. Lastly, the current study investigated only two variables of linear sprints (total distance and peak speed). In the future, researchers should expand to other variables and movement types concerning the utilizations of GNSS units in sports such as change in direction, acceleration, and deceleration.

## 5. Conclusions

In team sports, high consistency between different GNSS units is critical as coaches compare the movement characteristics across players in a game or training. This study revealed that there exist variations in distance and speed measurements among eight GNSS units worn by participants at the same time. In general, GNSS units may lead to an overestimation of total distance and underestimation of peak speed during high-speed sprint running. Practitioners should be aware of the window of errors associated with GNSS measurements and interpret the results with caution. This is especially important for data collected during sport competitions or training which involve movement demands at high speeds. When making comparisons over a season, players should wear the same GNSS unit every time if logistically possible.

Despite some limitations, the use of GNSS/GPS technology is still widespread, and it offers practical insights to players’ movements characteristics and playing demands. In view of the rapid advancement in technology, it may be possible to improve current GNSS/GPS systems so as to enhance their inter-unit consistency and measurement accuracies across different movement types including high-speed sprinting.

## Figures and Tables

**Figure 1 sensors-22-01888-f001:**
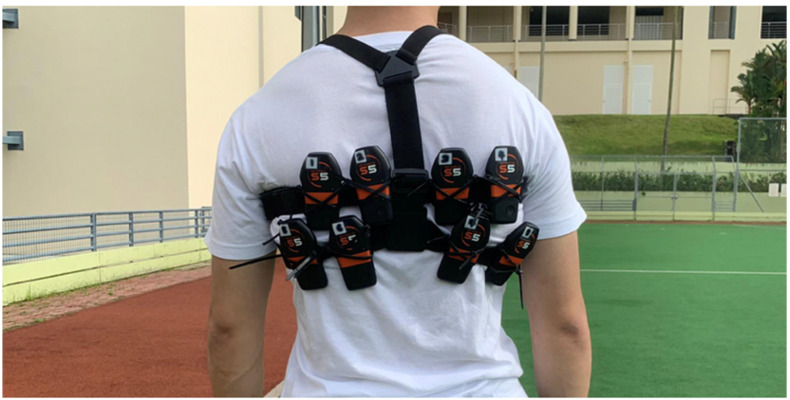
Each participant wore eight GNSS units at once in the maximal sprint test.

**Figure 2 sensors-22-01888-f002:**
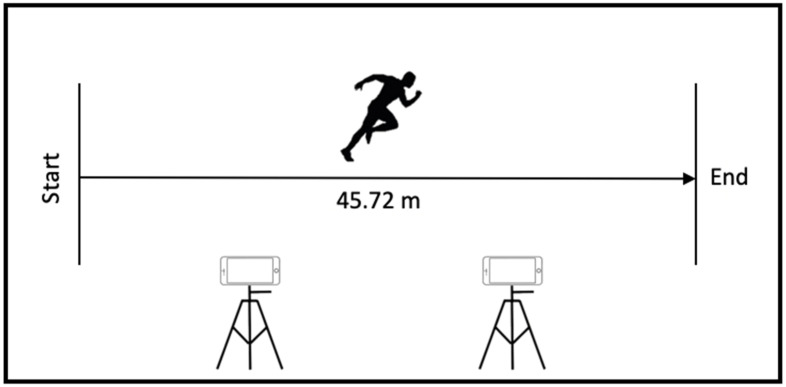
Experimental set-up of the sprint test over half a field hockey pitch (45.72 m) with two smartphone cameras recording the performances (Camera 1: 0 to 22.86 m, Camera 2: 22.86 m to 45.72 m).

**Figure 3 sensors-22-01888-f003:**
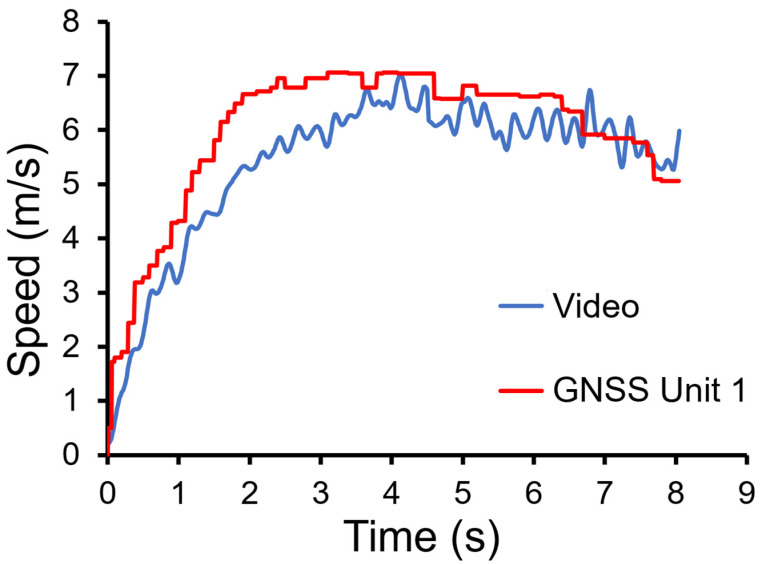
Representative raw speed-time from one participant measured using Kinovea video analysis and one GNSS unit.

**Table 1 sensors-22-01888-t001:** Reliability statistical outputs to assess inter-unit consistency.

GNSS Variables	ICC	95% Confidence Intervals	SEM
Total distance	0.131	[−0.024, 0.556]	8.8 m
Peak speed	0.323	[0.101, 0.736]	1.3 m/s

Note. ICC denotes intraclass correlation coefficients; SEM denotes standard error of measurement.

**Table 2 sensors-22-01888-t002:** Validity of GNSS distance measurements against known distance of 45.72 m.

GNSS Units	Mean (SD)	*p*-Value	Effect Size (*d*)	
Unit 1 (n = 13)	49.77 (5.92)	0.030 *	8.41	Large
Unit 2 (n = 13)	46.69 (10.62)	0.747	4.49	Large
Unit 3 (n = 10)	44.50 (8.55)	0.663	5.29	Large
Unit 4 (n = 13)	52.23 (10.11)	0.039 *	5.17	Large
Unit 5 (n = 11)	52.00 (10.13)	0.067	5.13	Large
Unit 6 (n = 12)	50.83 (8.57)	0.063	5.93	Large
Unit 7 (n = 12)	47.50 (8.06)	0.460	5.89	Large
Unit 8 (n = 13)	52.69 (12.18)	0.061	4.33	Large

Note. * Significant differences detected using one-sample *t*-tests (*p* < 0.05).

**Table 3 sensors-22-01888-t003:** Validity of GNSS peak speed measurements against video analysis.

GNSS Units	Mean (SD)	*p*-Value	Effect Size (*r*)	
Unit 1 (n = 13)	7.04 (1.15)	0.057	0.604	Large
Unit 2 (n = 13)	6.89 (1.98)	0.127	0.495	Medium
Unit 3 (n = 10)	6.92 (1.03)	0.064	0.673	Large
Unit 4 (n = 13)	7.11 (0.89)	**0.010 ***	0.780	Large
Unit 5 (n = 11)	7.37 (1.55)	0.416	0.303	Medium
Unit 6 (n = 12)	8.40 (2.53)	0.970	0.026	Negligible
Unit 7 (n = 12)	6.86 (1.34)	0.064	0.615	Large
Unit 8 (n = 13)	7.37 (1.05)	0.057	0.604	Large

Note. * Significant differences detected using Wilcoxon signed-rank tests (*p* < 0.05). Group mean (SD) of video-based peak speed was 7.78 (0.90) m/s.

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
