# Peer review of "Inter-Unit Consistency and Validity of 10-Hz GNSS Units in Straight-Line Sprint Running"

_sensors, 2022, doi:10.3390/s22051888_

Round 1
Reviewer 1 Report
With the development of GPS technology and sport gait data analyzing techniques, GPS has become a widespread and useful tool for both clinical practice and sport biomechanical research.
In this study video analysis was used as the golden standard for the speed measurement but references to the validation of the instrument are missing (L.133).
Validity was tested by comparing total distance and peak speed against criterion-referenced values (video-based reference values). I think it is very important to verify the GPS validity/variability, intraclass correlation coefficients (ICC) and Standard error of measurement (SEM) also in a series of straight-line sprint across the hockey pitch.
Reviewer 2 Report
The presented study analysis the inter-unit consistency and validity of 10-Hz GPS Units in Straight-line sprint running. The GPS-Units are evaluated by comparing the total distance traveled and peak speed observations to a camera based reference system. The paper concludes that there are a rather large inter-unit variations of up to 8.8 m and 1.3 m/s (SEM).
The paper is divided into four sections. After a comprehensive introduction, the author's present the material and methods applied for this research. Based on their experiments, statistical measures are applied and the inter-unit consistency is analyzed in section 3 while the results are discussed in section 4.
The collected data is analyzed carefully and supports the conclusions drawn by the author. There is no accuracy given for the reference system and some additional information in section 2 Materials and Methods is missing. The introduction can be also improved.
- Introduction:
- Please clarify "strength of the satellite configurations"
- Influence of SNR, satellite elevation, atmospheric effects, multi-path caused by surrounding objects are missing
- Please distinguish between GNSS and GPS, i.e. are different constellations used? Single or multi frequency bands?
- Materials and Methods
- Equipment: afaik Catapult S5 OptimEye uses GPS + Glonass, please clarify.
- Equipment: Are GNSS observations combined with inertial measurements? If yes, how?
- Equipment: Are the locations of the GPS-units affecting each other, i.e. compare SNR of different satellite signals for different receivers.
- Experimental protocol: is the actual running distance 45.72 m or longer, i.e. did the runners stop at the end-line? If they stopped at the end-line, why is the speed in figure 3 greater than 4 m/s at the end?
- Experimental protocol: Figure 2 indicates that smartphone cameras were used as reference, please confirm and clarify in the paper. Did you calibrate the camera system? Did you take optical effects such as lens distortion into account? What is the accuracy of your reference system?
- Experimental protocol: In order to receive the complete almanac, a startup time of 12.5 min is required. Can you guarantee same receiver conditions?
- Data processing: The number of experiments (13) is rather small as mentioned in 4.3.
- Data processing: please provide raw speed from videos
Round 2
Reviewer 1 Report
the paper has been improved and it is interesting in the field of GPS technology and sport gait data analyzing techniques
Author Response
Thank you for checking our revision and assuring the contribution of our work in the field of wearable technology and sport.

Reviewer 2 Report
You use a 10 Hz GNSS sensor and claim that no filtering is done internally. If no internal filtering is done, why are you sampling the device at 100 Hz?
Author Response
Thank you for raising this point. While our GNSS devices can sample at 100 Hz, data are exported at 10 Hz by default (referred as ‘GPS Rate’ in the software). Thus, the raw data we downloaded from the GNSS system were at 10 Hz and not 100 Hz. No down sampling is required. We apologize for the lack of clarity in our writing earlier and have since amended the manuscript to avoid confusion. The revised texts are:
“The GNSS movement data were downloaded using the manufacturer’s software (Catapult Sprint Version 5.1.7, Melbourne, Australia) at the default ‘GPS rate’ of 10 Hz.”
“Raw GNSS data were used without further down sampling, filtering, or smoothing procedures.”
